# Comparison between Carotid Distensibility-Based Vascular Age and Risk-Based Vascular Age in Middle-Aged Population Free of Cardiovascular Disease

**DOI:** 10.3390/jcm11164931

**Published:** 2022-08-22

**Authors:** Michaela Kozakova, Carmela Morizzo, Giuli Jamagidze, Dante Chiappino, Carlo Palombo

**Affiliations:** 1Department of Clinical and Experimental Medicine, University of Pisa, 56126 Pisa, Italy; 2Esaote SpA, 16152 Genova, Italy; 3Department of Surgical, Medical and Molecular Pathology and Critical Care Medicine, School of Medicine, University of Pisa, 56126 Pisa, Italy; 4Imaging Department, Fondazione Toscana G. Monasterio, 54038 Montignoso, Italy

**Keywords:** vascular age, risk factors, carotid distension, primary prevention

## Abstract

The concept of vascular age (VA) was proposed to provide patients with an understandable explanation of cardiovascular (CV) risk and to improve the performance of prediction models. The present study compared risk-based VA derived from Framingham Risk Score (FRS) and Systematic Coronary Risk Estimation (SCORE) models with value-based VA derived from the measurement of the common carotid artery (CCA) distensibility coefficient (DC), and it assessed the impact of DC-based VA on risk reclassification. In 528 middle-aged individuals apparently free of CV disease, DC was measured by radiofrequency-based arterial wall tracking that was previously utilised to establish sex- and age-specific reference values in a healthy population. DC-based VA represented the median value (50th percentile) for given sex in the reference population. FRS-based and SCORE-based VA was calculated as recommended. We observed a good agreement between DC-based and FRS-based VA, with a mean difference of 0.46 ± 12.2 years (*p* = 0.29), while the mean difference between DC-based and SCORE-based VA was higher (3.07 ± 12.7 years, *p* < 0.0001). When only nondiabetic individuals free of antihypertensive therapy were considered (n = 341), the mean difference dropped to 0.70 ± 12.8 years (*p* = 0.24). Substitution of chronological age with DC-based VA in FRS and SCORE models led to a reclassification of 28% and 49% of individuals, respectively, to the higher risk category. Our data suggest that the SCORE prediction model, in which diabetes and antihypertensive treatment are not considered, should be used as a screening tool only in healthy individuals. The use of VA derived from CCA distensibility measurements could improve the performance of risk prediction models, even that of the FRS model, as it might integrate risk prediction with additional risk factors participating in vascular ageing, unique to each individual. Prospective studies are needed to validate the role of DC-based VA in risk prediction.

## 1. Introduction

Atherothrombotic cardiovascular (CV) disease is the leading cause of morbidity and mortality worldwide, and, accordingly, effective and timely preventive interventions are required. The main goal of primary prevention is the identification of ‘high-risk’ individuals who would benefit from healthy lifestyle habits and more aggressive therapy. Constant adherence to medication and lifestyle interventions in high-risk individuals free of CV symptomatology depends in great part on motivation and, therefore, on an appropriate and effective explanation of the risk. The concept of vascular age (VA) was, therefore, introduced [1]. VA reflects the status of the vascular tree, and the comparison of individual VA with chronological age can provide a patient with a clear picture of risk.

There are two basic approaches for VA estimation, risk-based VA and value-based VA [2]. The first is based on existing risk models, as atherosclerotic and arteriosclerotic alterations of the vascular tree result from lifelong cumulative exposure to risk factors [3]. In this case, VA is calculated as the age of a person with the same predicted CV risk but with all risk factors within normal levels [4,5]. The second approach estimates VA from published sex- and age-specific percentiles of arterial wall thickness or stiffness obtained in healthy men and women [6,7,8,9].

Value-based VA has also been used to improve the predictive ability of risk models [10,11,12]. It is evident that degenerative changes in the arterial wall are caused not only by established CV risk factors but also by genetic predisposition, foetal programming and environmental factors. In fact, the existence of individuals with accelerated vascular ageing (early vascular ageing (EVA)) and individuals with delayed vascular ageing (supernormal vascular ageing (SUPERNOVA)) have been described [13,14]. Thus, replacing chronological age with VA derived from validated vascular biomarkers could incorporate into risk prediction algorithms additional risk factors participating in vascular ageing and unique to each person.

The aim of the present study was to compare risk-based VA derived from the two most frequently used risk models, the Framingham Risk Score (FRS) and the Systematic Coronary Risk Estimation (SCORE) model, with value-based VA derived from the measurement of common carotid artery (CCA) distensibility and also to assess the impact of carotid distensibility-derived VA on risk reclassification in a middle-aged population free of CV disease. The CCA distensibility coefficient (DC) was measured by the same radio-frequency-based device used to create sex- and age-specific reference values in a healthy population [8].

## 2. Materials and Methods

### 2.1. Study Population and Protocol

The study population consisted of 528 middle-aged individuals (45 to 65 years) free of overt CV disease referred for a primary prevention programme to the Clinic for Cardiometabolic Risk Prevention of the Department of Surgical and Medical Pathology, the University of Pisa, between December 2011 and January 2020. All individuals underwent an examination protocol that included medical history, anthropometry, brachial blood pressure (BP) measurements, a fasting blood test, ECG and a high-resolution carotid ultrasound. Diabetes mellitus was defined as fasting glucose ≥7.0 mmol/L or 2 h plasma glucose ≥11.1 mmol/L [15], hypercholesterolemia as total cholesterol >5.17 mmol/L and/or LDL-cholesterol >4.14 mmol/L and/or statin therapy, hypertriglyceridemia as fasting triglycerides >1.7 mmol/L and hypertension as systolic blood pressure >140 mmHg and/or diastolic blood pressure >90 mmHg [16].

### 2.2. Body Size and BP Measurement

Body weight (kg) and height (m) were measured, and body mass index (BMI, kg/m^2^) was calculated. Waist circumference (cm) was measured as the narrowest circumference between the lower rib margin and anterior superior iliac crest. Brachial BP was measured at two different visits by a validated digital electronic tensiometer (Omron, model 705cp, Kyoto, Japan) in participants seated for at least 10 min, using regular or large adult cuffs according to the arm circumference. At both visits, two measurements were taken, separated by 2 min intervals, and the average was calculated. The average of two separate visits was used to estimate BP (mmHg).

### 2.3. Calculation of Vascular Age Based on FRS

FRS-based VA calculation was performed as indicated in the paper describing the risk prediction model for the calculation of a 10-year risk of CV disease [4]. The algorithm considers age, total cholesterol, high-density lipoprotein (HDL) cholesterol, brachial systolic BP, ongoing treatment of hypertension, smoking and diabetes status and provides, besides the estimation of a 10-year risk, the estimation of VA. VA represents the age of a person with the same risk but with all other risk factors at a normal level (nontreated systolic blood pressure of 125 mm Hg, total cholesterol of 180 mg/dL, HDL of 45 mg/dL, nonsmoker, nondiabetic). The highest VA value calculated by the FRS prediction model is >80 years; in the statistical analysis of this study, individuals having a VA of >80 years were considered to have a VA of 80. The difference between FRS-based VA and chronological age was calculated (ΔAge FRS). The risk was classified as low, intermediate or high when the 10-year risk of CV disease was <10%, 10–20% or >20%, respectively.

### 2.4. Calculation of Vascular Age Based on SCORE

SCORE-based VA used the same principle as the FRS-based VA calculation, i.e., it indicates the age of the subject with the same CV risk but with risk factors within normal ranges [5]. The SCORE risk prediction model calculates a 10-year risk of fatal CV disease; the algorithm is based on age, sex, brachial systolic BP, total and HDL cholesterol and smoking status and is different for European regions with low and high CV risk. The low-risk chart was used for Italy [17]. Individuals having a VA of >80 years were considered to have a VA of 80. The difference between SCORE-based VA and chronological age was calculated (ΔAge SCORE). The risk was classified as low, intermediate or high when the 10-year risk of fatal CV disease was <2%, 2–5% or >5%, respectively.

### 2.5. Common Carotid Artery Distension Coefficient and Vascular Age

Measurement of CCA distension was performed in the afternoon, 3 h after a light meal, in a quiet room with a stable temperature of 22°, after resting comfortably for at least 15 min in the supine position. All individuals were asked to abstain from cigarette smoking, caffeine and alcohol consumption and vigorous physical activity for 24 h.

Carotid ultrasound was performed on the right CCA using an ultrasound scanner equipped with a 10 MHz linear probe (MyLab 70, Esaote, Genova, Italy) and implemented with a previously validated radiofrequency-based tracking of the arterial wall (QAS^®^) that allows an automatic and real-time determination of CCA diameter and distension with a high spatial and temporal resolution. Briefly, longitudinal images of the right CCA with a clear definition of both carotid walls were obtained, and a rectangular ROI was placed at the CCA segment starting approximately 1 cm before the flow divider. Arterial distension was measured in 32 scanning lines positioned within the ROI (sampling rate of 550 Hz on 32 lines). From the real-time distension curves, the diastolic carotid diameter and carotid distension were automatically measured and a distension coefficient (DC) was calculated as follows: DC = (ΔA/A)/pulse pressure, where A = π × (diastolic diameter/2)^2^, ΔA = π × [(diastolic diameter + Δdiameter)/2]^2^ − π × (diastolic diameter/2)^2^ and pulse pressure = systolic BP–diastolic BP [8]. Radiofrequency-derived measures represent an average over six consecutive cardiac beats. The mean of two acquisitions was used for statistical analysis. BP used in the calculation was measured at the left brachial artery (Omron, Kyoto, Japan) during each acquisition of the distension curves.

Intra- and interindividual variability of acquisitions was evaluated in 25 volunteers, including individuals with diabetes and hypertension. The acquisitions were performed twice, in two different sessions separated by 30 min, both by the same operator and by two different operators. Brachial PP was comparable between the different acquisitions (*p* = 0.88 and 0.69). Intra- and interindividual variability of CCA distension in two different acquisitions was expressed as a percentage of the absolute difference between the two acquisitions and was 7.5 ± 4.6 and 9.0 ± 6.9%, respectively [18].

DC-based VA was obtained in tables/nomograms reporting the sex- and age-specific percentiles of CCA DC measured by the same radiofrequency-based system in 3601 healthy men and women [8]. Age corresponding to the 50th percentile (median) of DC for given sex was considered a DC-based VA. The maximum VA reported in DC tables/nomograms is 80 years; the individuals with a DC lower than the median corresponding to the age of 80 (therefore, having a VA of >80 years) were considered to have a VA of 80. Individuals with extremely high and extremely low DC were identified as those with DC higher than the 95th percentile and lower than the 5th percentile for given sex and age, respectively [19].

### 2.6. Statistical Analysis

Data are expressed as mean ± SD, and categorical data as percentages. Variables with skewed distribution are summarised as median [interquartile range] and were logarithmically transformed for parametric statistical analysis. Wilcoxon test was used to test the mean difference between DC-based VA and chronological age or VA derived from risk models. To assess the associations between VA obtained by different approaches, Spearman correlation coefficient r was calculated. Multiple linear regression analysis with backward stepwise removal was used to identify the independent associations of DC with established risk factors used in prediction models. Statistical tests were two-sided, and significance was set at a value of *p* < 0.05. Statistical analysis was performed by JMP software, version 3.1 (SAS Institute Inc., Cary, NC, USA).

## 3. Results

Characteristics of the study population, values of CCA DC and values of VA based on risk models and on carotid distensibility are reported in Table 1.

The mean differences between DC-based VA and chronological age or risk-based VA are reported in Table 2. It is evident that DC-based VA was higher than chronological age. There was a good agreement between DC-based and FRS-based VA, with the mean difference being less than half a year. The mean difference between DC-based VA and SCORE-based VA was higher. However, when only nondiabetic individuals were considered (*n* = 410), the mean difference decreased to 1.35 ± 12.5 years (*p* = 0.01), and when individuals with ongoing hypertensive therapy were excluded, the mean difference in the remaining 341 individuals dropped to 0.70 ± 12.8 years (*p* = 0.24).

Replacement of chronological age by DC-based VA in the FRS model resulted in the reclassification of 28% of individuals into a higher risk category, and the percentage of individuals in the high-risk category increased from 26% to 42%. In the SCORE model, this replacement resulted in the reclassification of 49% of individuals into a higher risk category, and the percentage of individuals in the high-risk category increased from 10% to 48% (Table 2).

Table 3 compares the arithmetic difference between risk-based VA and chronological age (ΔAge) together with established risk factors between individuals with extremely high (DC above the 95th percentile of the reference population) or low DC (DC below the 5th percentile of the reference population) and those with DC in the 5th to 95th percentile. Individuals with extremely low DC had FRS-based VA significantly higher than chronological age, while individuals with extremely high DC had FRS-based VA lower than chronological age. The former also had higher BP, prevalence of antihypertensive treatment and T2DM and lower HDL cholesterol, while the latter had lower BP, prevalence of antihypertensive treatment and T2DM and higher HDL cholesterol. The difference in ΔAge between DC percentiles was less prominent but still significant when chronological age was subtracted from SCORE-based VA.

Table 4 reports the independent correlates of CCA DC. CCA DC was independently associated with age, systolic BP, HDL cholesterol, ongoing treatment for hypertension and diabetes mellitus, and these risk factors explained 43% of its variability. None of other possible risk factors (BMI, waist circumference, plasma glucose, LDL-cholesterol and triglycerides) entered the model.

## 4. Discussion

The identification of individuals with an increased risk of CV disease is a foundation of primary prevention. According to the Guidelines of the European Society of Cardiology, in asymptomatic men >40 years of age and women >50 years of age, a systematic or opportunistic evaluation of CV risk should be considered, and in individuals at intermediate–high risk, a healthy lifestyle strategy and preventive pharmacological treatment should be adopted [20]. Recommendations include a healthy dietary pattern with limited consumption of red meat, soft drinks and alcohol, weight control, smoking cessation, the substitution of sedentary behaviour with regular physical activity, strict control of T2DM and hypertension and, eventually, statin and aspirin therapy. Sustained adherence to these often unpopular recommendations depends on appropriate communication with the patient and a clear illustration of individual risk. For this reason, a theory of VA was adopted, assuming that the demonstration that one’s own arteries are older than chronological age is more convincing than a mathematical model calculating the chance of developing CV disease over the next 10 years [1,21].

VA can be estimated from risk prediction models (risk-based VA) as the age of a person with the same predicted CV risk but with all risk factors within normal ranges [4,5] or from vascular biomarkers of atherosclerosis and arteriosclerosis (value-based VA) as the age of a healthy person with the same value of measured vascular biomarker [2,6,7,8,9,22,23]. The most frequently used prediction algorithms are FRS and SCORE, and the most frequently used vascular biomarkers are carotid IMT and carotid–femoral pulse wave velocity (cfPWV) [10,11,12].

Arterial distensibility, in general, and carotid distensibility, in particular, have been proposed as possible biomarkers capable of improving CV risk prediction. A meta-analysis of nine longitudinal studies including 18 993 individuals has shown that carotid DC is a significant predictor of future CV events (pooled risk ratio 1.19 (1.06–1.35, 95%CI)) [24]. Therefore, in this study, we compared risk-based and DC-based VA and evaluated the impact of DC-based VA on risk reclassification in a large middle-aged population free of apparent CV disease but with various risk factors that may affect arterial compliance. We observed a good agreement between VA corresponding to the median value of DC in the healthy population and FRS-based VA. The mean difference was less than half a year. In contrast, the difference between DC-based and SCORE-based VA was 3 years (Table 2). This is not surprising, as the SCORE model does not take into account diabetes mellitus and antihypertensive treatment [17], which are important determinants of arterial stiffness and CV risk [25,26,27], and whose prevalence in our population was 22% and 23%, respectively. Indeed, when only nondiabetic individuals were considered, the mean difference decreased to 1.5 years, and when individuals with high BP treatment were also excluded, the mean difference dropped below 1 year. This observation indicates that the SCORE prediction model and SCORE-based VA should be used only in individuals free of diabetes and antihypertensive treatment, i.e., as a screening tool in an apparently healthy population.

A good agreement between DC-based and FRS-based VA reflects the fact that five out of seven risk factors considered in FRS [4] were independent determinants of DC, explaining 43% of DC variation (Table 4). The impact of established risk factors on carotid compliance was also evident in individuals with extremely low (EVA) or extremely high (SUPERNOVA) carotid compliance for their sex and age. The former had significantly higher BP, prevalence of diabetes and hypertensive treatment and lower HDL cholesterol as compared with individuals in the 5th to 95th percentile of the reference population, while the latter had the opposite trend. As a consequence, individuals with EVA had FRS-based VA much higher than chronological age (mean difference 15.8 ± 7.8 years), and individuals with SUPERNOVA had FRS-based VA lower than chronological age (mean difference −4.3 ± 8.7 years).

Numerous investigators have suggested substituting chronological age in the risk prediction model with VA derived from vascular biomarkers of atherosclerosis and arteriosclerosis in order to include in the risk prediction the factors that may participate in CV risk but are not clearly related to established risk factors, such as genetic predisposition, socioeconomic status, physical inactivity or psychological stress. In previous studies, the incorporation of value-based VA derived from sex- and age-specific IMT and cfPWV nomograms into risk models [10,11,12] resulted in the reclassification of 10–51% of individuals into a higher risk category [2]. In our population, the substitution of chronological age with DC-based VA resulted in the reclassification of 28% to a higher category of the FRS model and 49% to a higher category of the SCORE model. The reassignment of nearly half of the study population to a higher SCORE risk category once again indicates that the SCORE algorithm may underestimate risk when used in a population with diabetes and hypertensive treatment. Indeed, 111 out of 258 reclassified individuals had diabetes and/or hypertensive therapy.

Despite a good agreement between DC-based and FRS-based VA, more than a quarter of the population was reassigned to a higher FRS category when DC-based VA replaced chronological age in the FRS algorithm. Since the established risk factors explained only 43% of the DC variance, it is likely that other factors not accounted for in the FRS model, such as family history, habitual physical activity or dietary habits, could modify carotid compliance [28,29,30,31]. Thus, the incorporation of DC-based VA could integrate additional risk factors into the prediction algorithm. Most importantly, the substitution of chronological age with DC-based VA increased the prevalence of high-risk individuals, that is, those requiring more aggressive preventive interventions, in both models.

### Study Limitations

We did not use the latest SCORE risk prediction algorithm that calculates the 10-year risk of CV disease, as the VA for this new version was not yet established [32]. CV disease was excluded on the basis of clinical history and ECG; no provocative tests were performed. This was a cross-sectional study that did not allow to assess whether the reclassification with DC-based VA actually improved the prediction of CV events.

## 5. Conclusions

The present study indicates that the use of VA derived from the measurement of CCA distensibility might improve the performance of risk prediction models, especially the SCORE model, which does not account for the presence of diabetes and hypertensive treatment. Nevertheless, the replacement of chronological age by DC-based VA significantly increased the prevalence of high-risk individuals in the FRS model because the inclusion of VA could integrate risk prediction with additional risk factors unique to each individual. Prospective studies are needed to validate the true value of CCA DC-based VA for risk management in a population setting.

## Figures and Tables

**Table 1 jcm-11-04931-t001:** Characteristics of study population.

	Mean ± SD/Median [IQR]/n(%)
Gender M F	266 (50) 262 (50)
Age (years)	58.3 ± 5.5
BMI (kg/m^2^)	27.1 ± 4.7
Waist (cm)	96 ± 13
Systolic BP (mmHg)	132 ± 17
Diastolic BP (mmHg)	80 ± 10
Total cholesterol (mmol/L)	5.4 ± 0.9
HDL cholesterol (mmol/L)	1.6 ± 0.5
LDL cholesterol (mmol/L)	3.3 ± 0.8
Triglycerides (mmol/L)	1.1 [0.7]
Fasting glucose (mmol/L)	5.7 ± 1.4
Current smoker yes	116 (22)
Hypertension yes	152 (29)
Hypertension therapy yes	120 (23)
Hypercholesterolemia yes	297 (56)
Hypertriglyceridemia yes	106 (21)
T2DM yes	118 (22)
CCA DC (10^−3^ kPa^−1^)	14.0 ± 5.0
FRS-based VA (years)	65.5 ± 12.0
SCORE-based VA (years)	62.9 ± 7.9
CCA DC-based VA (years)	66.0 ± 13.8

**Table 2 jcm-11-04931-t002:** Mean difference and correlation between DC-based vascular age, chronological age and risk-based vascular age and reclassification of risk with DC-based vascular age.

				Reclassification n (%)	
	Mean Difference (Years)	*p*	Spearman r	↑ Risk Category	↓ Risk Category
**DC-based VA (years) vs.**					
Chronological age (years)	7.71 ± 13.4	*<0.0001*	0.26		
FRS-based VA (years)	0.46 ± 12.2	*0.29*	0.56	150 (28)	26 (5)
SCORE-based VA (years)	3.07 ± 12.7	*<0.0001*	0.42	258 (49)	32 (6)
**Risk Categories Low: Intermediate: High**	**Chronological Age; n (%)**			**DC-based VA; n (%)**	
FRS	219 (41):172 (33):137 (26)			177 (34):129 (24):222 (42)	
SCORE	281 (53):195 (37):52 (10)			180 (34):95 (18):253 (48)	

**Table 3 jcm-11-04931-t003:** Arithmetic difference between risk-based vascular age and chronological age (ΔAge) and established risk factors according to percentiles of carotid distension coefficient in reference population.

	CCA DC (10^−3^ kPa^−1^)		
	<5th Percentile	5–95th Percentile	>95th Percentile
N (%)	62 (12)	448 (85)	18 (3)
ΔAge FRS (years)	15.8 ± 7.8 **	6.5 ± 10.4	−4.3 ± 8.7 **
ΔAge SCORE (years)	7.7 ± 5.3 **	4.4 ± 4.6	0.7 ± 3.1 **
Systolic BP (mmHg)	146 ± 17 **	130 ± 15	111 ± 7 **
HDL cholesterol (mmol/L)	1.3 ± 0.4 **	1.6 ± 0.4	1.8 ± 0.6 **
Total cholesterol (mmol/L)	5.2 ± 0.9	5.4 ± 0.9	6.1 ± 0.8 *
Hypertensive therapy yes (n (%))	24 (39) **	96 (21)	0 **
Smoking yes (n (%))	16 (25)	98 (22)	2 (11)
T2DM yes (n (%))	33 (53) **	84 (19)	1 (6)*

Statistical significance tested against values in the 5th–95th percentile; * *p* < 0.05; ** *p* < 0.01–0.0001.

**Table 4 jcm-11-04931-t004:** Independent correlates of CCA distension coefficient.

	CCA DC (10^−3^ kPa^−1^)	
	β ± SE	*p*
Age (years)	−0.18 ± 0.03	*<0.0001*
Systolic BP (mmHg)	−0.48 ± 0.03	*<0.0001*
HDL cholesterol (mmol/L)	0.12 ± 0.03	*0.001*
Hypertensive treatment yes	−0.16 ± 0.04	*<0.0005*
Diabetes mellitus yes	−0.19 ± 0.04	*<0.0001*
*Cumulative R* ^2^	0.43	*<0.0001*

## Data Availability

The data presented in this study are available on request from the corresponding author. The data are not publicly available due to privacy reasons.

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
