# Peer review of "Comparison between Carotid Distensibility-Based Vascular Age and Risk-Based Vascular Age in Middle-Aged Population Free of Cardiovascular Disease"

_jcm, 2022, doi:10.3390/jcm11164931_

Round 1
Reviewer 1 Report
Your work undoubtedly arouses interest, because the analyzed topic and problem is current and its clinical importance is only growing. As a positive note, the differences in your concept indicated in the conclusion and discussion should be mentioned. The relatively small number of patients cannot be considered a significant drawback, because, at the same time, the working concept is definitely and will be useful as a reference for future research in this direction.
I recommend making the conclusion part more systematic and precise. Some sentences are more appropriate for the end of the discussion section.
Author Response
Thank you for your suggestion.
In the revised manuscript we tried to make the Conclusions more concise and we discuss some issues in details in the Discussion section.
Reviewer 2 Report
Kozakova et al in their article - Comparison Between Carotid Distensibility-Based Vascular Age and Risk-Based Vascular Age in Middle-Aged Population Free of Cardiovascular Disease - examine the differences between vascular age (VA) measured by risk factors and by the distensibility coefficient in 528 patients, respectively. They chose the two most widely used risk factor models, FRS and SCORE. They showed that especially the FRS-VA but also the SCORE, when the risk factors are not considered, have a good agreement with the DC-VA. When the chronological age is replaced by the CD-VA in these scores, this leads to a significant reclassification of patients into higher risk groups. This may be important for the prevention of vascular disease, so the authors raise an interesting and important issue here.
The study has the following minor weaknesses:
1. Please replace P for significance level with p and N for number of patients, etc. with n.
2. Please explain the numbers in brackets in table 3.
3. The last sentence on page 4, line 176/177 is duplicated and can be deleted.
In summary, Kozakova and colleagues address a very interesting and significant topic that is explained in a complicated way in some parts of the article.
Author Response
Thank you for your comments.
We performed the minor corrections required and we also tried to simplify the discussion.
Reviewer 3 Report
Dear authors
This study is well designed and the method is well enough. On the other hand the subject lacks of originality. Generally it is well presented. I want to ask to points:
Line 142-3: The aforementioned study was published from one of tha authors. Apparently you accept and use this method but please explain better this information and don't take it for granted.
Line 152-3: Please explain the thinking of this parameter. It is not clearly explained.
Maybe if you find a key point in the outcomes of this research, you could increase the interest.
Best regards
Author Response
Thank you for your suggestions.
We have explained in details the assessment of inter- and intra-individual variability (see page 6, second paragraph, lines 5-11).
We have explained better the concept of DC-based VA measurement and we hope that the meaning is now clear. Similarly to FRS-based and SCORE-based VA, the maximum age reported by DC nomograms is 80 years. So the subjects with DC lower than the median corresponding to 80 years were still considered as having VA of 80 years (see page 6, last 2 lines and page 7, first line).
We tried to express better the aim of the study and outcomes in Introduction and Discussion.
We have also performed English language use revision.
Round 2
Reviewer 3 Report
Dear authors
I think that you have made really substantial and useful changes to the manuscript. Good luck
Author Response
To improve the Methods section we add the paragraph describing in details the measurement of distension coefficient, on which is based the central message of the study. See page 6, paragraph 1.